# Atomic motifs govern the decoration of grain boundaries by interstitial solutes

Xuyang Zhou [1,2] ✉, Ali Ahmadian [2], Baptiste Gault [1,3], Colin Ophus[4], Christian H. Liebscher [2], Gerhard Dehm [2] & Dierk Raabe [1] ✉

Grain boundaries, the two-dimensional defects between differently oriented crystals, tend to preferentially attract solutes for segregation. Solute segregation has a significant effect on the mechanical and transport properties of materials. At the atomic level, however, the interplay of structure and composition of grain boundaries remains elusive, especially with respect to light interstitial solutes like B and C. Here, we use Fe alloyed with B and C to exploit the strong interdependence of interface structure and chemistry via charge-density imaging and atom probe tomography methods. Direct imaging and quantifying of light interstitial solutes at grain boundaries provide insight into decoration tendencies governed by atomic motifs. We find that even a change in the inclination of the grain boundary plane with identical misorientation impacts grain boundary composition and atomic arrangement. Thus, it is the smallest structural hierarchical level, the atomic motifs, that controls the most important chemical properties of the grain boundaries. This insight not only closes a missing link between the structure and chemical composition of such defects but also enables the targeted design and passivation of the chemical state of grain boundaries to free them from their role as entry gates for corrosion, hydrogen embrittlement, or mechanical failure.

Grain boundaries (GBs) are among the most important features of the microstructure, occupying an area of 500-1000 football fields per cubic meter material with 1 μm grain size. On the one hand, they render positive effects such as the simultaneous increase in strength and ductility[1,2], but on the other hand they make materials vulnerable, for crack initiation[3], onset of corrosion[4], or pinning of magnetic domain walls[5]. These features qualify GBs as the most important microstructural defects for many high-performance materials.

Two key factors are missing in our knowledge about GBs. The first one is the systematic and meaningful representation of GBs with pertinent and property-oriented parameters: their characterization requires five geometric degrees of freedom and further parameters to describe them down to their atomistic-scale facets[6,7] and atomic motifs[8-10], yet, it is still elusive which of these really matter for structure-property relations. By atomic motifs we mean here certain geometric polyhedra built by atoms that form repeating structural units to constitute the GBs[9,11,12], e.g., the kite[10], domino and pearl[8] structures that were described before to decipher the GB's elementary "genetic" structure. This concept of the atomic motif was inspired by the early works of Gaskell[13,14] and Bernal[15]: they described liquids and amorphous materials as an arrangement of polyhedral structural units and found that only a surprisingly small number of them is needed to construct such complex materials.

The second reason is the difficulty in characterizing both the GB structure and its chemical decoration state down to the atomic scale. For a systematic experimental study, we define five levels of hierarchy with respect to the crystallographic, compositional, and electronic features of GBs. The levels are (a) the macroscale interface alignment

[1]Department of Microstructure Physics & Alloy Design, Max-Planck-Institut für Eisenforschung GmbH, 40237 Düsseldorf, Germany. [2]Department of Structure & Nano- / Micromechanics of Materials, Max-Planck-Institut für Eisenforschung GmbH, 40237 Düsseldorf, Germany. [3]Department of Materials, Royal School of Mines, Imperial College London, SW7 2AZ London, UK. [4]National Center for Electron Microscopy, Molecular Foundry, Lawrence Berkeley National Laboratory, Berkeley, CA 94720, USA. ✉e-mail: x.zhou@mpie.de; d.raabe@mpie.de

and grain misorientation (held constant here); (b) the systematic mesoscopic change in the inclination of the GB plane for the same orientation difference; (c) the facets, atomic motifs, and internal nanoscopic defects within the boundary plane; (d) the GB chemistry; and (e) the electronic structure of the atomic motifs. Identifying key properties of GBs across all these scales at systematic variation of these parameters has not yet been realized.

The hierarchy of the GB structure across these different features is complex: an adequate representation of a GB requires at least to define the misorientation between adjacent grains and the inclination of the GB plane. At a mesoscopic view energy-favored GB planes are normally the most densely packed ones which comprise a high density of coincidence sites, derived from a virtual lattice that extends over both adjacent crystals[16]. On a more refined scale, we find that the GB planes do not generally follow this geometrical criterion, but can further decompose into a sequence of piecewise planar facets that are sets of lower total energy reconstruction motifs[17]. At the atomic level, local GB structures, including the faceting[6,17], as well as many other features such as secondary defects[18], ordering[19−21] and phase transformation[8,22], can play crucial roles in the spatial distribution of solute atoms within GB planes. This complex interplay between GB structure and chemistry leads to a deviation from the Langmuir-McLean type of (sub-)mono-layer adsorption behavior[23], which takes a thermodynamic view at chemical decoration, yet, is structurally agnostic to these fine atomic details. This complexity shows that—in order to gain a fundamental understanding of the coupling behavior between structure and che-mical composition—direct experimental observation of all these structure features conducted across all these scales, together with a mapping of the GBs chemical composition is required. This must be done under conditions, where each of these parameters is varied sys-tematically, step-by-step, while the other parameters are kept constant.

For this study, we have developed a custom-designed workflow along an instrument ensemble which allows to map the structure, chemistry and electronic state of GBs at length scales ranging from macroscopic to atomic. As a model material we have chosen a Bridgeman-produced body-centered cubic (bcc) – Fe Σ5 (where Σ5 stands for the density of coincident sites, denoted as the coincident site lattice (CSL) value[16]) bicrystal, stabilized by approximately 4 at.% Al and alloyed with C and B atoms[24]. We have selected this material as Fe-C alloys, also referred to as steels, represent about 1.9 billion tons of material produced each year[25], by far the most common metal class, with an uncounted number of safety critical and functional applica-tions, ranging from huge infrastructure to tiny magnets. B has been selected as a second alloy element owing to its peculiar, and often highly beneficial effect on GB cohesion and the resulting material properties. At the smallest scale, we are able to reconstruct the charge-density maps from the differential phase contrast – four dimensional scanning transmission electron microscopy (DPC-4DSTEM) data[26−28] to directly spatially resolve light (low atomic number) interstitial atoms decorating the atomic motifs of GBs. With this multi-scale and multi-physics GB analysis approach we make the surprising observation that it is actually not the macroscopic or mesoscopic geometrical aspects that explain the specific chemical decoration state of an interface as is usually claimed. Instead, features at the smallest length scales, i.e. the atomistic motifs, determine the chemistry of a GB. Analogous to building blocks or Lego bricks, a limited number of atomic motifs with their sequential alignment form complex GB structures with a wide range of microscopic degrees of freedom and varying solubility for accommodating solute atoms.

## Results and discussion
### Hierarchical characterization of grain boundaries
We grew bcc−Fe Σ5 bicrystals[24]. The macro- and mesoscale study of GB structure was carried out by optical, scanning electron microcopy (SEM), and diffraction imaging. During the growth process, the GB plane changed its orientation with respect to the neighboring grains, see the orange dashed line superimposed on the optical image (Fig. 1a). We cut a disk (5 mm thick and 2 cm in diameter, marked as a blue rectangle in Fig. 1a) from the bicrystal to investigate the hier-archical structure and composition of the GB over 9 orders of mag-nitude in size scale using a variety of characterization methods. Macroscopically, this particular region exhibits a significant change in the GB plane inclination perpendicular to the growth direction, but remains relatively straight along the growth direction, see the optical image (Fig. 1a).

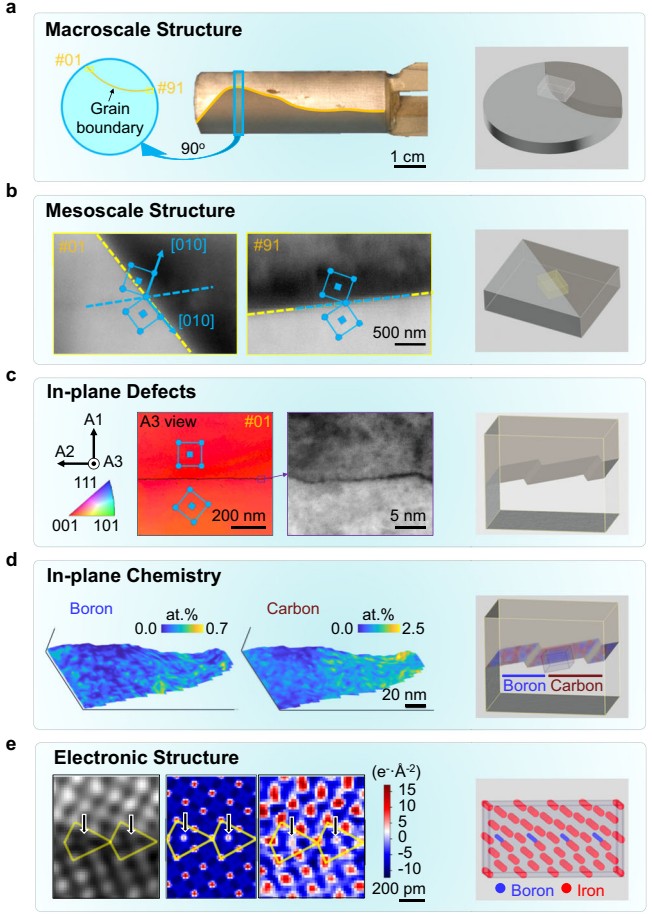

**Fig. 1 | Hierarchy study of Σ5 Fe grain boundaries (GBs) over 9 orders of mag-nitude. a** Macroscale information of the Σ5 bicrystals. **b** Structural characterization of Σ5 GBs at mesoscale level by electron backscatter diffraction (EBSD). A con-tinuous change in the inclination of the GB was recorded while the misorientation between the adjacent grains remained constant. The number indicates the position at which the EBSD scan was acquired, ranging from #1 - # 91, at 10 μm spacing, see the illustration in a. Here only two representative regions are shown. The blue squares represent the body-centered cubic (bcc) unit cells imaged at [001] direc-tion. **c** shows in-plane defects in the representative Σ5 (430) // (010) GB. The left-hand side images show the reconstruction of the orientation from the four-dimensional scanning transmission electron microscopy (4DSTEM) data set. The blue squares represent the body-centered cubic (bcc) unit cells imaged at [001] direction. The right image is the high angle annular dark field (HAADF)−scanning transmission electron microscopy (STEM) image. **d** The local composition of B and C along the Σ5 (430) // (010) GB plane quantified from atom probe tomography (APT) data set. **e** Imaging light B and C atoms in the center of the kite structure for a Σ5 (310) // (3̄10) GB using differential phase contrast (DPC)-4DSTEM imaging. The image includes the reconstructed dark-field (the first image from left) and charge-density maps for both simulated (second from left) and experimental (third from left) data sets. The last column of each row shows the schematic illustration of the main structural features of a given hierarchy.

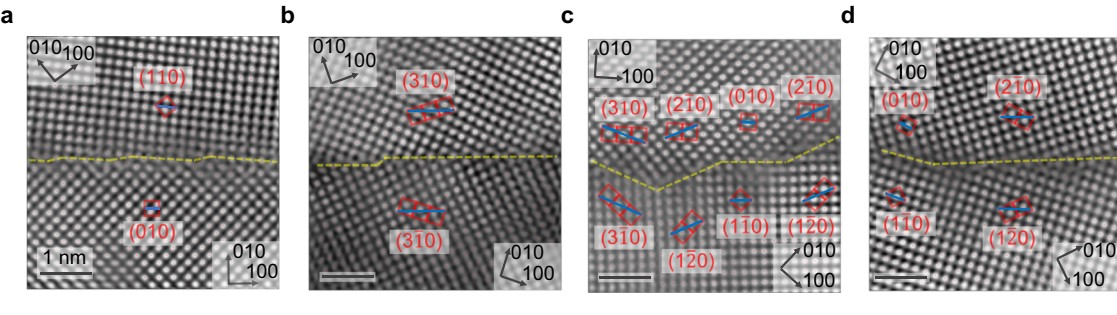

**Fig. 2 | Local facetted GB structures. a–d** are Σ5 GBs with the mesoscale plane pairs (430) // (010), (310) // (3$\bar{1}$0), (11 1 0) // (9$\bar{8}$0) and (2$\bar{1}$0) // (1$\bar{2}$0), respectively. The yellow lines indicate the positions of the GBs. The red squares and blue lines assist in identifying the local GB planes. **e** Summary table showing the local GB planes for Σ5 GBs with different mesoscopic GB inclinations.

| ROI | Macroscopic grain boundary planes | Local grain boundary planes | | | | |
|---|---|---|---|---|---|---|
| | | Type 1 | Type 2 | Type 3 | Type 4 | Type 5 |
| #01 | (430) // (010) | (110) // (010) | (310) // (3$\bar{1}$0) | (210) // (4$\bar{1}$0) | (520) // (4$\bar{1}$0) | - |
| #41 | (310) // (3$\bar{1}$0) | (310) // (3$\bar{1}$0) | (110) // (010) | - | - | - |
| #71 | (11 1 0) // (9$\bar{8}$0) | (010) // (1$\bar{1}$0) | (310) // (3$\bar{1}$0) | (3$\bar{1}$0) // (1$\bar{3}$0) | (2$\bar{1}$0) // (1$\bar{2}$0) | (410) // (2$\bar{1}$0) |
| #91 | (2$\bar{1}$0) // (1$\bar{2}$0) | (2$\bar{1}$0) // (1$\bar{2}$0) | (1$\bar{1}$0) // (100) | (010) // (1$\bar{1}$0) | (1$\bar{3}$0) // (130) | (3$\bar{1}$0) // (2$\bar{3}$0) |

Figure 1b shows the mesoscale GB character measured by electron backscatter diffraction (EBSD). The GB has a total length of approximately 1 cm. We divided the GB into regions of interest (ROIs) from #1 to #91 with 10 μm horizontal distance between each number. The orientation of the adjacent grains remained constant, while the GB plane systematically changed its angle, see the inset blue cubes and yellow lines. We quantified the rotation of the GB planes by the inclination, which is the angle between the GB plane and the inner bisector between the [010] directions in the grains. The inclination changed from 0° to 60° with a monotonic variation (more measurements are referred to Supplementary Fig. 1).

At the nano- and atomic scales, we found that GBs are not flat. Faceting, discontinuities, and steps occur at a high number density along the GB plane. For example, the 4DSTEM orientation map and the high-angle annular dark-field−scanning transmission electron microscopy (HAADF- STEM) image (Fig. 1c) show facet structures along the Σ5 [430]//[010] GB. Such abrupt local structural variation leads to a dramatic change in the spatial distribution of solute atoms within the GB planes, showing that it is not the mesoscopic alignment but the atomic-scale motif- and facet-structure which determines its chemical decoration. Figure 1d shows a representative atom probe tomography (APT) measurement of the same Σ5 [430]//[010] GB, where the local content of C and B varies significantly along the curved GB plane. We present more 4DSTEM orientation maps in Supplementary Fig. 2 and APT reconstructions in Supplementary Fig. 3.

Using the recently developed DPC-4DSTEM imaging technique, we directly observed the light B and C atomic columns at the local atomic motifs of GBs[27,28]. As shown in Fig. 1e, the atomic columns containing the B or C atoms at Σ5 (310) // (3$\bar{1}$0) GB were revealed and validated by a combination of experimental and simulated charge-density maps reconstructed from the DPC-4DSTEM data sets. The contrast for the light atoms, here B or C, is weak in the dark-field image due to their much lower atomic weight compared to Fe. Using the charge-density maps, we were able to detect a clear signal in the center of the GB atomic motif structure (referred to as "kite"[10,17]), representing the B or C-containing atomic columns. The charge-density map can also provide an opportunity to study the electronic structure of the GB[28]. Here we focused on applying the charge-density map to the direct imaging of light atoms at the local atomic motifs of Fe GBs in the electron microscope with sub-Ångström-resolution.

## Facet structures

We have systematically quantified the GB structures for a series of Σ5-GBs with different mesoscale inclinations using HAADF-STEM images. The analyses include two symmetric GBs, #41 Σ5 (310) // (3$\bar{1}$0) GB and #91 Σ5 (2$\bar{1}$0) // (1$\bar{2}$0) GB, as well as two asymmetric GBs, #01 Σ5 (430) // (010) GB and #71 Σ5 (11 1 0) // (9$\bar{8}$0). The four representative GBs are selected and displayed in Fig. 2a–d. More images of these GBs can be found in Supplementary Fig. 4. We have drawn yellow lines to highlight the projection of GB planes where atomically sharp GBs were detected. This indicates that the lamellae samples are so thin that the inclination of the GBs in the projection direction is not pronounced, i.e., the structures are essentially constant along the projection direction.

Figure 2a shows the Σ5 (430) // (010) GB in which in-plane defects occur continuously to compensate for the lattice mismatch between the (110) and (010) planes among the adjacent grains. The Σ5 (310) // (3$\bar{1}$0) GB appears relatively straight, with the local atomic motifs showing the typical kite structure (Fig. 2b). In this GB, a step connecting two regions with kite structure was observed. The formation of this step is attributed to the fact that the current bicrystal deviates by 1.1° from the exact misorientation required for the perfect formation of a Σ5 GB[29]. For the Σ5 (11 1 0) // (9$\bar{8}$0) GB (Fig. 2c), we found a nano-facet morphology with different combinations between local atomic motifs, such as (310) // (3$\bar{1}$0), (2$\bar{1}$0) // (1$\bar{2}$0), (010) // (1$\bar{1}$0), etc. The Frank-Bilby equation[30] suggests that long-range coherency strains resulting from the misorientation (1.1° deviation from the true Σ5 GB) and inclination can be canceled out by Burgers vectors existing at different junctions of the facetted GB structures. For example, it has been reported by Medlin et al. that an array of (1/5)[310] and (1/5)[120] dislocations can fully accommodate the misorientation and inclination from the true Σ5 GB[17]. The dislocations that exist at the junction of the facetted structures play a key role in reducing the coherency strain between the two adjacent regions and stabilizing the GB structure. In the last case of the Σ5 (2$\bar{1}$0) // (1$\bar{2}$0) GB (Fig. 2d), the majority of the local atomic motifs are the straight (2$\bar{1}$0) // (1$\bar{2}$0) motifs with the intervening (010) // (1$\bar{1}$0) facet. We tabulated all observed local atomic motifs (Fig. 2e), where three specific ones dominate, namely

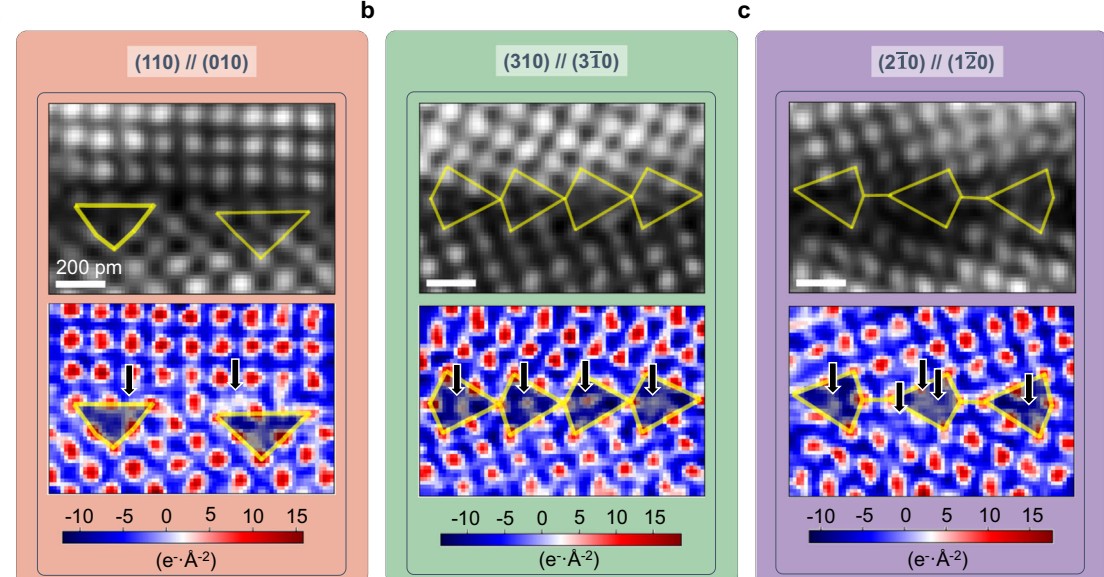

**Fig. 3 | Imaging light B and C atoms at atomic motifs.** Three representative local atomic motifs: (**a**) (110) // (010); (**b**) (310) // (3̄10); (**c**) (2̄10) // (1̄20) For each local atomic motif, the dark-field and charge-density map reconstructed from the atomic DPC-4DSTEM data sets are shown in the top and bottom panels, respectively. The repeated structures are highlighted in yellow lines on the dark-field map. The B or C atomic columns are indicated by black arrows in the charge-density maps.

{110} // {010}, {310} // {310}, and {210} // {120}. For the symmetric Σ5 atomic motifs, including {310} // {310} and {210} // {120}, the kite-shaped structural units are identified and also predicted from Mendelev and density functional theory (DFT) calculations[17,31]. However, there are less reports on the asymmetric {110} // {010} atomic motifs.

## Imaging light B and C atoms at atomic motifs

We have next studied the above mentioned three types of prevalent atomic motifs with regards to the occurrence of the light elements B and C, using the DPC-4DSTEM method[26–28]. B and C atomic columns are identified by locating peaks with weak or no contrast in the dark-field images, but strong contrast in the charge-density maps. The full reconstruction, including the electric field, the (projected) electro-static potential, and the charge-density maps, can be found in Supplementary Fig. 5. Here we only show the reconstructed dark-field image (top row) and the charge-density map (bottom row) for each atomic motif (Fig. 3a–c). More ROIs of the local atomic motifs can be found in Supplementary Fig. 6. It is worth noting that this DPC-4DSTEM method can only detect whether there is an atomic column consisting of light elements but it cannot distinguish what type of solute it is, i.e. whether it is a predominantly B or C-occupied atomic column or a mixed state (see Supplementary Fig. 7). All of these occupation states have been shown to be energetically favorable for (310) // (3̄10) atomic motifs based on our preliminary DFT study[24].

In Fig. 3a, we highlighted the (110) // (010) atomic motif with a yellow triangle. The B and C atomic columns are marked with black arrows in the charge-density map, which appear far apart, indicating a potentially low solute content in this type of atomic motif. The interstitial atoms tend to occupy the (distorted) tetrahedral sites at the atomic motifs, e.g. the right black arrow in Fig. 3a. Figure 3b shows the typical kite structure of the (310) // (3̄10) atomic motif, highlighted by yellow quads. The charge-density map shows that the center of the kite structure is occupied by closely spaced atomic columns, indicating a higher segregation tendency than in the (110) // (010) atomic motif. Under the theoretical framework of the structural unit model[9,11,12,32,33], the Σ5 (310) // (3̄10) GB is composed of nesting (distorted) capped trigonal prisms with the interstitial atoms in the center. Such interstitial sites occur only in regions of defects, such as dislocations or GBs, and are fundamentally different from the tetrahedral or octahedral

sites in the bulk bcc structure[9]. The large coordination number and long bond distance give these interstitial sites a higher preference for segregation of interstitial atoms[32].

The last (2̄10) // (1̄20) atomic motif is more complicated, as we identified two different variants of this type. Figure 3c shows one variant of the atomic motif that appears to be a separated kite structure, as highlighted with yellow quads and a line. Similar to the (310) // (3̄10) atomic motif, the B or C atomic columns can also occupy the centers of the kite structure (the capped trigonal prims) highlighted by the first and third arrows from left in Fig. 3c. Moreover, in this type of atomic motif, there is a new type of structural unit, namely the pentagonal bipyramid[32], with its center highlighted by the second arrow from the left in Fig. 3c. Another variant of this atomic motif for the Σ5 (2̄10) // (1̄20) type structure is the extended kite structure (Supplementary Fig. 6c), which has also been reported from DFT calculations[34].

## Grain boundary chemistry

We have selected the same four representative GBs for the in-plane GB chemistry analysis using APT. To better understand the influence of the GB inclination on their chemical properties, we selected ROIs in which the GBs appeared to be flat in the reconstructed APT volume. For example, Fig. 4a shows the atom maps of Fe, B, and C, extracted from a region with a Σ5 (310) // (3̄10) GB. We found a clear tendency for B and C to segregate at the GB, which is consistent with previous theoretical and experimental studies[24,32,35]. It is worth noting that the detailed study of Al depletion caused by the co-segregation of B and C was reported in our previous publication[24]. In the current work, we focus on the segregation behavior of the light elements B and C.

We quantified the in-plane solute compositions of B and C throughout the GB (see Fig. 4b). The composition ranges for B and C are 0.2–1.3 atomic (at.) % and 0.2–1.7 at. %, respectively, corresponding to interfacial excesses of 0.7–4.5 atoms (at.)·nm⁻² and 0–5.7 at.·nm⁻², respectively. In a GB composed of kite structures, the center of a single kite structure includes several sites in the direction perpendicular to the projection direction of the kite structure; these sites may be fully or partially occupied by interstitial atoms, resulting in different solute content, i.e., different values of interfacial excess. When the center of the kite structure consisting of a Σ5 (310) // (3̄10) GB is fully occupied

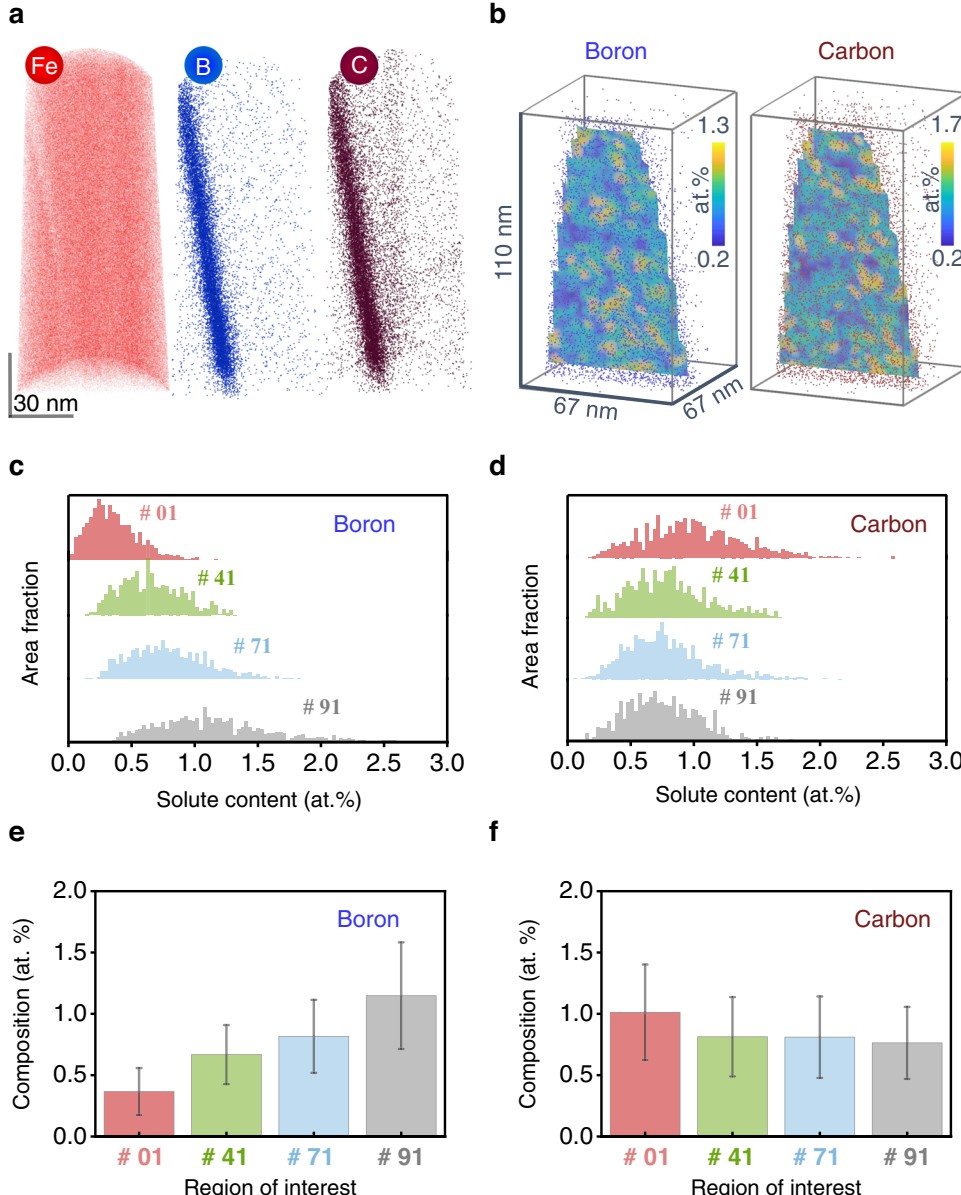

**Fig. 4 | Local chemical composition analysis for Σ5 GBs. a** Fe (only show 3 at. % of total Fe atoms), B, and C atom maps from the APT reconstruction of the Σ5 (310) // (3$\bar{1}$0) GB. **b** Contour maps of the solute B and C compositions along the GB planes are shown in a. The distribution of (**c**) B and (**d**) C atoms along the GB planes of #01 Σ5 (430) // (010), #41 Σ5 (310) // (3$\bar{1}$0), #71 Σ5 (11 1 0) // (9$\bar{8}$0) and #91 Σ5 (2$\bar{1}$0) // (1$\bar{2}$0) GBs, represented by the area fraction of a given B or C content as a function of B or C composition. **e** and **f** are the average compositions of B and C for GBs shown in c and d. The error bars indicate the standard deviations.

by interstitial atoms, the interfacial excess is 7.7 at.·nm$^{-2}$. Our APT results show there is less than a monolayer adsorption of B or C atoms at this Σ5 (310) // (3$\bar{1}$0) GB, since the interfacial excesses for both B and C are less than 7.7 at.·nm$^{-2}$. The mean occupancies for B and C are approximately 31% and 38%, respectively.

We also noticed an inhomogeneous segregation pattern of both B and C in the contour maps of their compositions in the GB planes[36] (see Fig. 4b). First, it must be clarified that due to the limited detection rate and trajectory aberration[37], APT cannot provide a reconstruction in which every atom in the bulk material is accurately reconstructed. Here for the spatially resolved solute content along the GB plane, the quantification results represent the statistics over several, approximately 10–20, kite structures. The solute content values in Fig. 4b do not show the composition of each individual kite structure, but the average value for several kite structures. Although the composition of each kite provides more information about the physical

interrelationships from the energy point of view. It is not possible to directly learn the atomic level information from the APT measurements. However, the statistics of an average value can readily provide a general understanding of the behavior of solute-solute interaction at GBs[38,39].

Interestingly, the decoration features with B and C are not strongly correlated but some regions appears to be mutually repulsive, i.e., regions enriched in B can have a low content of C (see Fig. 4b) and vice versa. We quantified the Pearson product-moment correlation coefficient between the local GB compositions of B and C as 0.02, which is midway between the lower (−1, anti-correlated) and upper (1, perfect correlation) bounds, indicating a slightly weak correlation. There are two main reasons for the inhomogeneous segregation. The first reason is composition-dependent segregation behavior. In our previous work, we used DFT to study this phenomenon and we found that increased coverage of interstitial or substitutional sites, e.g. by B

and C, decreases the segregation tendency, suggesting that co-segregation effects limit the enrichment of B and C[24]. More specifically, the segregation energy for B at the interstitial sites is −2.6 eV · −2.8 eV, indicating a strong segregation tendency. However, when additional B or C is present at the second nearest (to GB) interstitial sites, the segregation energy for interstitial sites approaches zero or even reaches positive values. In these cases, further segregation is energetically unfavorable[24].

The second reason is site- and motif-specific segregation, which is determined by the local atomic motifs in the GB plane[6,32,33]. Wang et al.[32] found that more open GB structures, e.g., with local atomic motifs such as (310) // (3$\bar{1}$0) or (2$\bar{1}$0) // (1$\bar{2}$0), are energetically more favorable for C segregation compared to more compact structure units, such as e.g. the local (2$\bar{1}\bar{1}$) // (1$\bar{2}$1) atomic motif. The tetrahedral sites observed for the (110) // (010) atomic motif are more compact than the centers of the capped trigonal prisms or the pentagonal bipyramids for the (310) // (3$\bar{1}$0) and (2$\bar{1}$0) // (1$\bar{2}$0) atomic motifs, indicating a lower tendency for solute segregation. The atomic motif-dependent segregation tendency has also been reported for the system Cu-Ag by Peter et al.[6], where nanometer-sized facets were observed to be composed of preferentially Ag-segregated symmetric Σ5{210}//{210} segments and Ag-depleted {230}// {100} asymmetric segments. In our study, the inhomogeneous B and C segregation is therefore attributed to the changes in local atomic motifs.

Additional in-plane quantitative GB chemistry results are summarized (Fig. 4c–f, each of the APT reconstruction in the Supplementary Fig. 3). The average B content increased from 0.37 ± 0.19 at. % to 1.15 ± 0.43 at. % as the GB inclination decreased from 45° for Σ5 (310)//(3$\bar{1}$0) to 0° for Σ5 (2$\bar{1}$0) // (1$\bar{2}$0). The B segregation at the Σ5 (2$\bar{1}$0) // (1$\bar{2}$0) GB is approximately twice that at the Σ5 (310)//(3$\bar{1}$0) GB and more than three times that at the Σ5 (430) // (010) GB. For the average in-plane composition of C, a relatively constant composition of 0.8 at. % was observed for GBs with different inclinations, except for the Σ5 (430) // (010) GB, which is about 25% higher than for the other GBs. Summarizing the segregation of B and C, a maximum solute content of 1.95 at% is observed for the Σ5 (2$\bar{1}$0) // (1$\bar{2}$0) GB (Supplementary Fig. 8). Lejcek and Hofmann have previously reported the anisotropic GB segregation behavior of P, Si, and C at various tilt GBs of Σ = 5, 36.9° [100] in an Fe-3.5 at.%Si alloy using Auger electron spectroscopy[40,41]. Special (singular) segregation behavior has been reported not only for symmetrical GBs, such as {013} and {012}, but also for asymmetrical GBs, e.g., (001)/(03$\bar{4}$), (0$\bar{1}$7)/(01$\bar{1}$), and (001)//(011)[42]. The authors attribute the singular segregation behavior of these GBs to the high values of the interplanar spacing[41].

The wide composition range (Fig. 4d) and high average content of solutes for the Σ5 (2$\bar{1}$0) // (1$\bar{2}$0) GB are due to the following reasons. Two atomic motifs have been reported to constitute the local Σ5 (2$\bar{1}$0) // (1$\bar{2}$0) GB structure from DFT predictions, either a separated kite or an extended kite structure[33,34]. Wang et al. indicates that the atomic rearrangement from the extended kite structure to the separated kite structure results in a reduction of GB energy by 0.44 J·m$^{-2}$ [32]. In this work, we observed both types of local atomic motifs from the previous DPC-4DSTEM analysis (Fig. 3c, Supplementary Fig. 6c). The variance in local atomic motifs and energy for the same GB structure provide varied preference for the segregation sites, causing a wider composition range. The segregation energies for C at the interstitial site reduced from −1.13 eV · atom$^{-1}$ to −1.40 eV · atom$^{-1}$ when the atomic motif changes from (310) // (3$\bar{1}$0) to (2$\bar{1}$0) // (1$\bar{2}$0) at full coverage[32], indicating that the latter is energetically more favorable for C segregation than the former. In addition, we frequently found that the (2$\bar{1}$0) // (1$\bar{2}$0) atomic motifs are commonly more distorted and can accommodate higher fractions of interstitial or substitutional sites that are suited for hosting B or C (Supplementary Fig. 6c). The direct observation of atomic motif structures underneath the mesoscale plane inclination features provide an explanation for the profound differences between GBs with different inclinations for their chemical decoration states observed by APT.

The variation in GB composition arises from atomic motif types and the nano-facetted structure observed for the different types and degrees of GB plane inclination. We had picked and presented three typical atomic motifs for direct visualization of light solute atoms B and C, as shown in Fig. 3 and Supplementary Fig. 6. The (110)//(010) atomic motif has fewer interstitial sites, while the (2$\bar{1}$0) // (1$\bar{2}$0) motif can accommodate more solute atoms by varying and accommodating its local motif structure and the associated distortions. The (310) // (3$\bar{1}$0) atomic motifs are in-between these two types, specifically regarding their solubility to accommodate and host solute atoms. Combining these atomic motifs along the GB plane in the form of a motif sequence thus leads to variations in the local GB chemical composition. For instance, the GB with an inclination of 60° is primarily formed by low solubility atomic motifs, such as (110) // (010) and (310) // (3$\bar{1}$0), which translates to the lowest overall solute content for this specific GB. Since the GB with 0° plane inclination contains a high fraction of (2$\bar{1}$0) // (1$\bar{2}$0) type atomic motifs, it also exhibits the highest overall solute content. Through high-resolution atomic structure analysis and detailed chemical composition analysis, we discovered the strong dependence of the GB's solute segregation behavior on its specific underlying atomic motif types and their sequential alignment into a facetted arrangement. The resulting atomic structure affects the ability of GBs to accommodate solute atoms. With this we can trace a GB's segregation behavior back to the solubility and arrangement of its underlying atomic motifs. This changes our view of GB segregation from a top-down approach, where segregation has been associated with macroscopic kinematic degrees of freedom to a bottom-up approach, where the nature of the atomic motifs that are the underlying interfacial building blocks as well as their sequential arrangement determine a GB's solubility.

In summary we performed a holistic, multiscale analysis of the hierarchical, structural, and chemical features of GBs using high-resolution imaging techniques including HAADF-STEM, DPC-4DSTEM, and APT, which extends over 9 orders of magnitude in spatial resolution. The state-of-the-art charge-density maps directly resolved the atomic columns of light elements B and C at the GBs of an Fe bicrystal. We find that the chemical properties of GBs cannot be simply determined by their CSL value or mesoscopic angular features alone, but instead strongly depend on their GB inclination, the resulting faceting, and particularly on the local atomic motifs. These results not only yield direct experimental evidence for understanding the chemical nature of GBs on the basis of their atomic-scale structural properties, but also provide the scientific basis for developing advanced materials with controllable interfaces. Building upon the knowledge gained from this study, we plan to expand our investigation by exploring the temperature-dependent behavior of solute segregation and its influence on various mechanical and physical properties, such as corrosion, hydrogen embrittlement, and mechanical failure. These efforts will contribute to a more comprehensive understanding of the behavior of GBs and facilitate the development of more robust and reliable materials.

## Methods

### Bicrystal growth and sample preparation

The presented GB structures were obtained from growing bicrystals of an Fe-Al-B-C alloy with a Σ5 GB. An in-house modified Bridgman technique was employed for the sample preparation, in which two seeds were aligned on their common [001] axis and rotated symmetrically with a misorientation of Θ = 38° perpendicular to their common axis. The bi-crystal was grown at a temperature of 1600 °C, with a growth rate of 2 mm · hour$^{-1}$ for 50 hours. Following the growth process, the oven was slowly cooled down over a period of 24 h at a rate of approximately 1.1 °C•minute$^{-1}$. This choice of a slow cooling rate

aimed to minimize non-equilibrium segregation effects[24,43]. During the heat treatment, the temperature continued to decrease, and the resulting segregation state was dependent on the overall history of the cooling process. In our previous publication[24], we developed a kinetic model to determine the kinetic limitation in terms of a lower threshold temperature for the segregation of B, Al, and C, with 1200 K, 800 K, and 390 K, respectively, below which the segregation was kinetically limited. The enrichment of solutes is predominantly caused by the binding energy of the solutes with the GB[24]. Further details on the bicrystal synthesis and heat treatment can be found in our previous publication[24]. It is worth mentioning that the wet chemical analysis of the bulk sample showed a content of 4 at.% Al, 0.001 at.% B, and 0.05 at.% C. The addition of Al serves to stabilize the bcc-Fe phase.

For this work, we cut a disk (5 mm thick and 2 cm in diameter) from the original part of the bicrystal (see Fig. 1). The disk was first polished to obtain a mirror-like surface. Using an FEI Helios Nanolab 600i focused ion beam (FIB) dual beam microscope, we milled a series of marks to determine the location of the GBs. For this sample, a total length of approximately 1 cm GB was found, divided into ROIs from #01 to #91, separated by a horizontal distance of 10 μm between each number (see Supplementary Fig. 1a).

We used transmission electron microscopy (TEM) and APT to analyze the GB structure and composition of four selected ROIs, namely #01, #41, #71, and #91. The in-plane lift-out method (electron beam in TEM perpendicular to the common axis [001] of the grains) was used to prepare the TEM lamellae. First, the Cu grid was mounted on the correlative holder and placed horizontally in the FIB chamber for the lift-out process. Then, the chamber was vented for rotating the correlative holder 90° to thin the sample. The TEM lamellae were initially thinned to less than 60 nm at an accelerating voltage of 30 kV and then cleanly polished to a thickness of less than 30 nm at an accelerating voltage of 5 kV. The APT tips were fabricated using the conventional FIB lift-out method[37]. The wedge extracted by the FIB was mounted on a Si coupon and sharpened at 30 kV into needle-like geometries required for field evaporation, with a subsequent 5 kV cleaning step to remove surface damage implanted with Ga⁺.

## Orientation mapping

Information on GB character was obtained by EBSD analysis (see Supplementary Fig. 1b) of 9 ROIs (#01-#91) of the polished surface of the bicrystal disk using a JEOL-JSM-6490 microscope operated at 30 kV and equipped with an EDAX/TSL EBSD system. In addition, we performed mapping of the grain orientation and GB characters for the TEM lamellae using precession assistant 4DSTEM[44] (see Supplementary Fig. 2). 4DSTEM data sets have been acquired using the TemCam-XF416 pixelated complementary metal-oxide-semiconductor detector (TVIPS) in a JEM-2200FS TEM (JEOL) operated at 200 kV. During the data acquisition, the incident electron beam was precessed by 0.5° to create a quasi-kinematic diffraction pattern and scanned with a step size of 2.5 nm. The collected diffraction patterns of 4DSTEM data set were indexed by ASTAR INDEX program and the orientation was mapped using an offline TSL OIM Analysis 8 software package.

## Atom probe tomography (APT)

In this work, the quantification of GB chemistry was primarily conducted by APT. As for GB chemistry, the composition of a GB can also be measured by energy dispersive X-ray spectroscopy or electron energy loss spectroscopy. However, the use of these two techniques to study the segregation of B and C at the Fe GBs is not very promising for the following reason: The content of B and C is normally quite low, usually <2 at. % (see Fig. 4).

The GB composition and element distributions of the bicrystals were characterized by APT performed in a Cameca Instruments Local Electrode Atom Probe (LEAP) 5000 XS operated with a specimen set

point of 40 K and a laser pulse energy of 30 pJ at a pulse repetition rate of 200 kHz for a 0.5% atoms per pulse detection rate. The collected data sets were reconstructed using the AP Suite 6.1 software platform (Supplementary Fig. 3). A calibration procedure was applied to obtain the correct image compression factor and k-factor for generating the proper shape and lattice spacing for the reconstructed volume[45].

The overall compositions for the reconstructed volume of the tips extracted from ROIs #01, #41, #71, and #91 are $Fe_{95.47}Al_{4.34}B_{0.04}C_{0.14}$ (at.%), $Fe_{95.38}Al_{4.45}B_{0.07}C_{0.11}$, $Fe_{95.37}Al_{4.38}B_{0.11}C_{0.14}$, $Fe_{95.31}Al_{4.42}B_{0.14}C_{0.14}$, respectively. Here, the peak decomposition algorithm must be applied to quantify the bulk composition because there is significant peak overlapping between $Al^+$ and $Fe^{2+}$ at 27 Da (measured in Daltons, mass-to-charge ratio). The peak decomposition algorithm is only suitable for statistical compositional analyses. It does not provide spatial resolution, e.g., distinguishing compositional differences between two 1 nm³ cubes 1 nm apart. If we want to quantify the one-dimensional (1D) compositional profile across the GB, we need to assign the 27 Da to either $Al^+$ or $Fe^{2+}$ ions. In the 1D composition profile of the Supplementary Fig. 9, the 27 Da peak was assigned to the $Fe^{2+}$ ion, resulting in a significantly lower Al content than the value obtained from the wet chemical analysis or the peak decomposition analysis. A more detailed investigation of the Al depletion caused by the co-segregation of B and C can be found in our previous publication[24]. In the present work, we have mainly focused on the segregation behavior of B and C atoms. It is also worth noting that the local B and C contents obtained from APT composition quantification are significantly higher than the results from the global wet chemical analysis, which is due to GB segregation and a high fraction of GB contained in the reconstructed tip volume.

We used the APT_GB software[36] to quantify the in-plane chemical distribution of B and C atoms at the GB, i.e., the composition map and the interfacial excess map (Supplementary Fig. 3). The GB planes were identified by a pre-trained convolutional neural network[36] and meshed triangularly with a unit size of approximately 8 nm². Ladder diagrams were calculated for each vertex of the mesh to determine the interfacial excess and local composition[36].

The correlation between the local GB compositions of B and C is evaluated using a weighted Pearson product-moment correlation coefficient. A pair of variables $(C_B, C_C)$ represent a pair of B and C compositions of the nodes in the GB meshes. We obtain these composition values from the in-plane chemical distribution described in the previous paragraph. The third variable $w$ corresponds to the area of the nodes. The weighted correlation coefficient is written as

$$\text{corr}(C_B, C_C; w) = \frac{\text{cov}(C_B, C_C; w)}{\sqrt{\text{cov}(C_B, C_B; w)\text{cov}(C_C, C_C; w)}} \quad (1)$$

where weighted covariance $\text{cov}(C_B, C_C, w)$ can be calculated as follow

$$\text{cov}(C_B, C_C; w) = \frac{\sum_i w_i \cdot (C_{Bi} - m(C_B; w))(C_{Ci} - m(C_C; w))}{\sum_i w_i} \quad (2)$$

here $m(C_B; w)$ is the weighted mean:

$$m(C_B; w) = \frac{\sum_i w_i C_{Bi}}{\sum_i w_i} \quad (3)$$

## High angle annular dark field - Scanning transmission electron microscopy (HAADF-STEM)

All high-resolution HAADF-STEM data were acquired using a Cs probe-corrected FEI Titan Themis 60-300 (Thermo Fisher Scientific) equipped with a high-brightness field emission gun and a gun monochromator operating at 300 kV. Images were recorded with a HAADF detector (Fishione Instruments Model 3000) at a probe current of

68 pA using a semi-convergence angle of 23.6 mrad. The semi-collection angle for high-resolution HAADF-STEM images was set to 103-220 mrad. Image series of at least 20 images were acquired with a dwell time of 2 μs at a pixel size of 6 pm. To minimize the effects of instrumental instabilities in the images, we used non-rigid registration and averaging of image series to achieve sub-picometer precision measurement of atomic column positions in high-resolution HAADF-STEM images[46]. The stacked images were also processed with the Bragg filter and the double Gaussian (band-pass) filter to remove background noise. Additional high-resolution HAADF-STEM images of ROIs #01, #41, #71 and #91 are shown in Supplementary Fig. 4, illustrating the diversity of local GB facet structures. We have also provided the raw stacked images in Supplementary Fig. 10. The boundary planes for these GBs are perpendicular to the paper, resulting in clear imaging conditions for the atomic columns on both sides of the boundary. However, this imaging condition is not always satisfied. Some HAADF-STEM images of selected ROIs are shown in Supplementary Fig. 11, where the top grain overlaps with the bottom grain in the direction perpendicular to the paper.

## Atomic four-dimensional STEM (4DSTEM) data collection

The atomic 4DSTEM data were also acquired in the Titan microscope at 300 kV. We collected the entire convergent beam electron diffraction (CBED) pattern as a two-dimensional (2D) image for each probe position during scanning. These images were taken using an electron microscope pixel array detector (EMPAD) with a readout speed of 0.86 ms per frame and a linear electron response of 1,000,000:1. Each CBED image has a size of $128 \times 128$ pixel$^2$. All data sets were acquired with a semi-convergence angle of 23.6 mrad, a defocus value of approximately 0 nm, and a camera length of 300 mm. The exposure time was 1 ms per frame. Beam scanning was synchronized with the EMPAD camera with a scanning step size of 18 pm and a field of view of $2.3 \times 2.3$ nm$^2$. The scanning step size was optimized by comparing the reconstructed 4DSTEM data sets with step sizes of 13-36 pm. The selection took into account the possibility of minimizing distortions due to instrument instability and maximizing spatial resolution to resolve light atoms. We calibrated the CBED pattern in reciprocal space using the standard Au nanoparticle. Here, each pixel in the CBED pattern is 2.0 mrad.

## Electron dose quantification for 4DSTEM data acquisition

Irradiation of materials with accelerated electrons can initiate ballistic knock-on processes that lead to displacement of atoms from the crystal lattice and produce point defects, e.g., a Frenkel pair consisting of an interstitial and a vacancy[47]. For Fe, the maximum transferable kinetic energy at 300 kV is 15.25 eV, which is slightly lower than the displacement energy of 16.00 eV[47]. Theoretically, the radiation damage should not be significant when imaging Fe materials. When acquiring atomic 4DSTEM data sets for imaging light atoms at GBs, a lower dose is preferred.

We quantified the dose with the Gatan camera for energy-filtered TEM (EFTEM) by varying the beam via defocusing a monochromator. The measured dose has been shown as a function of the defocus of the monochromator (Supplementary Fig. 12). The red and blue dots show that the doses for 55 and 95 monochromator defocuses are $5.3 \times 10^5$ $e^- \cdot ^{-2}$ and $1.9 \times 10^5$ $e^- \cdot ^{-2}$, respectively. These two values were used to acquire the atomic 4DSTEM data sets. We did not observe any significant changes in the reconstruction results recorded with these two doses.

## Atomic 4DSTEM data reconstruction for experimental data

The originally collected atomic 4DSTEM data contains the 2D grid of the probe position in real space and the 2D diffraction pattern for each probe position in reciprocal space[48]. Data reconstruction is required to obtain information such as the (virtual annular) dark-field image, the

electric field map, the (projected) electrostatic potential map, and charge-density map[27] (see Supplementary Fig. 5). The python script pyDPC4D was developed for data reconstruction (GitHub link: https://github.com/RhettZhou/pyDPC4D)[49]. The script is forked from the py4DSTEM package[50]. We used py4DSTEM to reconstruct the dark-field image and to calculate center of mass (c.m.) of the transmitted beam for each probe position. Our pyDPC4D script mainly focused on the quantitative reconstruction of the electric field map, the (projected) electrostatic potential map, and the charge-density map. The details of the reconstruction are as follows.

**Dark field image.** The dark-field images are reconstructed from 4DSTEM data sets by integrating the intensity of the annular region of 99-122 mrad in the CEBD pattern of each probe position. Supplementary Fig. 5a shows an example of the reconstructed dark-field image containing (110) // (010) atomic motifs.

**Electric field map.** Atomic electric fields can be measured using DPC-4DSTEM microscopy, an imaging technique that reflects the relative electron probe shifts observed on CBED patterns due to local electric and magnetic fields[26-28,51-54]. This method can resolve the structure of weakly interacting phase objects. For example, when an electron probe passes through an electric field, the electron is deflected due to its negative charge. By quantifying the shifts of the transmitted electron probe in the diffraction plane ($\langle \triangle d^* \rangle$, shift of the c.m.), the change in momentum of the electron probe ($\mathbf{P}_\perp$) can be calculated. With appropriate modeling, the electric field of the materials under study ($\mathbf{E}_\perp$) can also be derived. In the simplest model of a uniform electric field, the momentum transfer of the electron is negatively proportional to the electric field. In classical electrodynamics, the electric field is equal to the Lorentz force divided by the charge, which equals the momentum transfer of electrons per time and per charge. According to the Ehrenfest theorem[55], this also holds in quantum mechanics[27]. Considering the weak phase object approximation[56] for thin samples, the electron beams move through the sample without changing their velocity in the z direction. The following equation represents the relationship between the momentum transfer ($\mathbf{P}_\perp$, or shift of the c.m., $\langle \triangle d^* \rangle$) and the measured electric field $\mathbf{E}_\perp$:

$$\mathbf{E}_\perp = -\langle \mathbf{P}_\perp \rangle \frac{v}{et} = -\left\langle \triangle d^* \right\rangle \frac{hv}{et} \tag{4}$$

here, $v = 2.33 \times 10^8 \, m \cdot s^{-1}$ is the speed of electrons at 300 kV, $e = 1.6022 \times 10^{-19}$ C is the elementary charge, $t \approx 15.0 \times 10^{-9} \, m$ is the thickness of the specimen, and $h = 6.6261 \times 10^{-34} J \cdot s$ is the Planck's constant.

We applied a circular mask (radius 74 mrad) to the CEBD pattern to calculate the c.m. The purpose of applying a mask is twofold. First, it can eliminate intensity from the high-angle scattering that is less sensitive to the momentum transfer than that from the low-angle scattering. Second, it can also reduce the noise from c.m. calculation. Supplementary Fig. 5b shows the c.m. of the electron beam in the vertical (left image) and the horizontal (right image) directions. The electric field can be further calculated according to Eq. (4) (see Supplementary Fig. 5c for the electric field magnitude).

**(Projected) electrostatic potential.** We calculated the (projected) electrostatic potential ($\phi(\mathbf{r})$) by integrating the electric field in the plane perpendicular to the electron beam using the following equation.

$$\phi(\mathbf{r}) = -\int_{\mathbf{r}_1}^{\mathbf{r}_2} \mathbf{E}_\perp \cdot d\mathbf{r} \tag{5}$$

Here, $\mathbf{E}_\perp$ is the electric field obtained in the previous step. $r$ is the integrating distance in the real space. We performed the integration by using Fourier transform with the adding of low- and high- pass regularization terms to minimize noise[50]. Supplementary Fig. 5d presents the reconstruction of the (projected) electrostatic potential of the region containing (110) // (010) atomic motifs.

**Charge-density map.** The charge-density can be derived by calculating the divergence of the measured electric field, since they are proportionally correlated according to Gauss's law[27,28,53], see the following equation:

$$\rho = \varepsilon_0 \mathrm{div}\, \mathbf{E}_\perp = -\varepsilon_0 \mathrm{div}\langle \mathbf{P}_\perp \rangle \frac{v}{et} = -\frac{\varepsilon_0 hv}{et}\mathrm{div}\langle \triangle d^* \rangle \tag{6}$$

where $\varepsilon_0 = 8.8542 \times 10^{-12} C \cdot V^{-1} \cdot m^{-1}$ is the permittivity of the vacuum. Assuming that the charge-density is uniform along the z-direction, the (projected) charge-density (number of $e^-$ per area) can be written as:

$$\rho^{N-2D} = -\frac{\varepsilon_0 hv}{e^2}\mathrm{div}\langle \triangle d^* \rangle \tag{7}$$

Supplementary Fig. 5e shows the charge-density map for the (110) // (010) atomic motifs. We found that light atoms can be well resolved in the reconstructed charge-density map. Therefore, in this work, we will mainly show the charge-density map for imaging light atoms at GBs.

More examples for three representative atomic motifs, (110) // (010), (310) // ($3\bar{1}0$), and ($2\bar{1}0$) // ($1\bar{2}0$), are shown in Supplementary Fig. 6, in which light atoms are pointed by black arrows in the charge-density map. For these atomic columns, there is a weak contrast in the dark-field image but a relatively strong contrast in the charge-density map. By comparing GBs with the same misorientation and inclination, different local atomic motifs can be clearly resolved, for instance, Supplementary Fig. 6a i vs ii for the (110) // (010) atomic motif, and Supplementary Fig. 6c i-ii vs iii-vi for the ($2\bar{1}0$) // ($1\bar{2}0$) atomic motif. In addition, defects, such as disconnections and steps, can also strongly affect the local atomic motif and the location of atoms, e.g., Supplementary Fig. 6b iv.

**STEM multi-slice image simulation**
The purpose for conducting image simulation is twofold. First, we want to estimate the specimen thickness by comparing the experimental and simulated CEBD patterns. Second, the image simulations serve as an important tool to help interpret our experimental results, in particular to validate atomic columns observed by the reconstructed charge-density map. STEM multi-slice simulations were performed using the μSTEM (v5.2) package[57]. We took the (310) // ($3\bar{1}0$) atomic motif as the model structure for the image simulations. The atomic structure model was based on our DFT study[24], see Supplementary Fig. 13a, where the red atoms are Fe and the blue atoms are B. The microscope parameters, such as semi-convergence angle (23.6 mard), primary electron energy (300 kV), and scanning pixel size (18 pm) were identical to the atomic 4DSTEM experimental values. We used the μSTEM package to generate simulated 4DSTEM data sets that have the same format as the experimental data. The rest of the reconstruction was performed using the in-house developed python script pyDPC4D[49].

**Measurement of sample thickness**
CBED was acquired to determine the thickness of the TEM sample, because the thickness of the TEM sample has been shown to have a strong effect on the CEBD pattern[28,58]. Here we have examined some experimental CEBD patterns and compared them with the simulated

CEBD patterns. Supplementary Fig. 13b shows three of the experimental CEBD patterns extracted from the 4DSTEM data set with their positions highlighted in purple, orange, and green in the atomic model of the Σ5 (310) // ($3\bar{1}0$) GB (see Supplementary Fig. 13a). In the rightmost column of each row, we plotted the position-averaged CBED (PACBED) patterns. Furthermore, we present the reconstructed charge-density maps and the CBED patterns from the same regions for the simulated 4DSTEM data of the Σ5 (310) // ($3\bar{1}0$) GB with thicknesses from 4.9 nm to 20.1 nm (see Supplementary Fig. 13c). The thickness of the TEM sample was determined to be 14.9 nm because the experimental PACEBD patterns at such a thickness best match the simulated PACEBD patterns. In addition, we found that the contrast of the atomic columns on the charge-density map became complex patterns instead of a simple circular shape when the sample is thicker than approximately 17 nm. The simulations were performed up to a thickness of 40 nm (only shown up to 20 nm). An example of the complex charge-density map can be found in the reconstruction of the 20 nm thick model structure (see Supplementary Fig. 13c vi).

**Atomic 4DSTEM data reconstruction for simulated data**
We performed three series of simulations to understand the origin of the contrast appearing in the charge-density map. The benchmark simulation (see Supplementary Fig. 13c v) was performed with a sample thickness of 14.9 nm and a defocus of 0 nm for the atomic structure shown in Supplementary Fig. 13a.

In the first series, we reconstructed the charge-density maps by systematically changing the defocus from −100 nm to 80 nm (see Supplementary Fig. 7a i-vi). When the defocus is positive, the contrast in the reconstructed charge-density map reversed and appeared as complex patterns (see Supplementary Fig. 7a v-vi). We found it difficult to interpret such reconstruction results. However, the reconstructions produced with negative defocus generally provided good contrast to resolve each of the atomic columns (see Supplementary Fig. 7a i-iii).

The main elements in the bicrystals are Fe, Al, B, and C. In the second series, we investigated the influence of the solute type and structure on the contrast of the charge-density map. The simulated results include interstitial and substitutional B and C sites (see Supplementary Fig. 7b i-iv), Fe sites in the center of the kite structure (see Supplementary Fig. 7b v), and interstitial Al sites (see Supplementary Fig. 7b vi). No difference is apparent in charge-density maps reconstructed with B- or C-segregation, regardless of whether they are interstitial or substitutional sites. The DPC-STEM method does not provide mass resolution to distinguish these two light elements. For chemical information, please refer to our APT quantifications. It is worth noting that when the B or C atoms occupy the substitutional sites, a clear contrast can be seen between these atomic columns and the other columns without substitutional sites. We can frequently detect such a contrast difference in the experimental charge-density maps (see Fig. 3 and Supplementary Fig. 6), indicating a possible substitutional occupancy at GBs.

When light B and C atoms occupy the interstitial sites in the center of the kite structure, the contrast in the HAADF image is very weak. The most popular atomic motif for the Σ5 (310) // ($3\bar{1}0$) GB is the unfilled kite structure, see the HAADF-STEM and dark field images in Figs. 2b and 3b, Supplementary Fig. 4b, and Supplementary Fig. 6b. The atomic columns of B and C are only visible in the charge-density maps. However, in some cases we can see the contrast in the center of the kite in both the dark field images and the charge-density maps. In these cases, we believe that this contrast comes from the Fe occupation in the kite center, see Supplementary Fig. 7b v. For comparison, we also simulated the charge-density map where Al remains in the kite center as an interstitial site. We can also observe a contrast in the charge-density map, but it appears diffuse, see

Supplementary Fig. 7b vi. In the real case, the Al interstitial sites rarely occur for two main reasons. In our previous work, we found that the Al atoms preferentially occupy the substitution sites instead of the interstitial sites[24]. Second, there is an Al depletion induced by co-segregation of C and B[24]. See also the APT results in the Supplementary Fig. 9.

In the last series, we investigated the influence of atom occupancy on the contrast of the charge-density maps (see Supplementary Fig. 7c). We constructed the GB atomic models containing two B atomic columns: one has 100% atom occupancy, serving as a reference, and the other with a range of atom occupancies between 0% and 100%. It was found that the contrast of the partially occupied atomic column is proportional to the percentage of atom occupancy.

## Data availability

The experimental data generated in this study have been deposited in the public community repository Figshare (https://doi.org/10.6084/m9.figshare.22722916).

## Code availability

The custom python scripts "pyDPC4D" for reconstruction and analysis of DPC-4DSTEM data is available on GitHub. Link: https://github.com/RhettZhou/pyDPC4D[49].

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

## Acknowledgements

X.Z. acknowledges the support from Alexander von Humboldt Foundation (X.Z.). X.Z. acknowledge funding by the German Research Foundation (DFG) for funding via project HE 7225/11-1 (X.Z.). Work at the Molecular Foundry was supported by the Office of Science, Office of Basic Energy Sciences, of the U.S. Department of Energy under Contract No. DE-AC02-05CH11231 (C.O.).

## Author contributions

D.R. and G.D. conceived of the presented idea. X.Z. conducted the experiments and analytical characterization. A.A. provided bicrystalline sample. C.H.L. and C.O. guided transmission electron microscopy analysis. B.G. guided atom probe analysis. All authors provided critical feedback and helped shape the research, analysis, and manuscript.

## Funding

## Competing interests

The authors declare no competing interests.
