## [Peer Review File · Nature Communications]

Atomic motifs govern the decoration of grain boundaries by interstitial solutesREVIEWER COMMENTS

Reviewer #1 (Remarks to the Author):

This contribution represents a detail HRTEM study of the structures of the tilt grain boundaries in nearly $\Sigma 5$ CSL orientation relationship documented by the APT measurements of the grain boundary chemistry in bcc-based iron system. As a starting material, the authors used a sequence of the samples cut of a curved grain boundary in a bicrystal. In this respect this work would represent an important contribution to understanding the structure/property relationship and hint for the grain boundary engineering approach. From this point of view, the paper represents an excellent contribution to the field.

Unfortunately, there are some problems which degrade the quality of the top experimental approach and do not allow me to recommend the manuscript for publication in the present form. There are two main reasons supporting my decision:

I. The paper completely neglects the previous experimental results which were obtained earlier with the same conclusions despite more simple microscopic methods were used:

In 1990s, the group of Lejcek studied (in numerous papers) the segregation of Si, P and C in bicrystals of the bcc Fe-Si based alloy with the conclusion showing that (310) and (210) symmetric boundaries and (001)//(011) asymmetric boundaries are singular. In fact, this is the same conclusion as done in the present manuscript. In a paper, Lejcek et al. did also use the same geometry of the sample as the authors in the present research. Nevertheless, no paper of this group (except the book (23) but in a quite different context) is cited in the manuscript.

II. The interpretation of the GB segregation is vague.

As one can understand from the description in Methods, the samples were cut from the as-grown bicrystal. So, the reader understands that the segregation state cannot be ascribed to defined temperature. This is also supported by the width of the segregation layer (fig. 9/Extended data). It means that one cannot make ANY conclusion about the segregation behavior of individual grain boundaries! It is not clear whether the GB composition can correspond to low temperatures at which the segregation is high at general boundaries compared to singular GBs (= facets) or at high temperatures where the behavior is opposite? In addition, the composition of the grain boundaries in cooled as-grown bicrystal is out of any equilibrium, so no conclusion concerning the different behavior of individual GBs can be made from the composition of the boundaries.

III. Further important comments:

1. The grown bicrystal in this work contained two grains inclined by 38° around [001] axis and is ascribed to $\Sigma 5$ CSL orientation relationship. However, the true $\Sigma 5$ CSL misorientation is 36.9° [001]. If the study is done with the atomic resolution, the effect of the 1.1° deviation on the GB structure should be well explained. Very probably, this deviation can be explained by array of GB dislocations... Question is,

however, how the complex structures are dependent on such deviation, e.g., that of the (11 1 0)//(980) GB?

2. When speaking about the nano-faceting, the representation $\Sigma 5$ (010)//(110) is rather unlucky. I understand that it is the facet in $\Sigma 5$ orientation relationship but in fact, (010)//(110) interface is incommensurable, i.e., non-CSL one as these two planes differ by $\sqrt{2}$ which is irrational number. I would recommend to use only (010)//(110) without using ' $\Sigma 5$ '.

3. L. 240: 'repulsive interaction'... The behavior of B and C is definitively more complicated. First, there will be size competition as these two solutes are interstitial. Repulsion is definitively not excluded but I do not know the details. Additionally, the interaction of these solutes with Al can play a role.

4. L. 265: 'the GB inclination decreased from 60° for (310)//(310) to 0° for (210)//(210).' (310) and (210) symmetric GBs are inclined by 45° . Can you explain, what does mean these ' 60° '?

Reviewer #2 (Remarks to the Author):

The authors performed a holistic, multiscale analysis of the hierarchical, structural, and chemical features of GBs using high-resolution imaging techniques including HAADF-STEM, DPC-4DSTEM, and APT. They find that the chemical properties of GBs cannot be simply determined by their CSL value or mesoscopic angular features alone, but instead strongly depend on their GB inclination, the resulting faceting, and particularly on the local atomic motifs. Some suggestions are shown as follow:

1. The authors exploit the strong interdependence of the interface structure and chemistry and find that the change of the inclination of the grain boundary plane with identical misorientation impacts grain boundary composition and atomic arrangement. What is the relationship between the inclination of GB, type of GB and GB compositions.

2. I was expected more investigations on this topic. The authors have shown the microstructure of GBs including the inclinations of GB, GB composition, and atomic arrangement, but how these elements influence the mechanical and transport properties of materials (such as corrosion, hydrogen embrittlement, or mechanical failure) is more important? Without this part of work, this manuscript looks uncompleted.

3. Fig. 2 can be divided into two figures (a-e for fig.2 for Facet structures discussion, f-h for fig.3 for Imaging light B and C atoms at atomic motifs discussion).

RE: NCOMMS-22-53810

Atomic motifs govern the decoration of grain boundaries by interstitial solutes

Author response to reviewer comments

REVIEWER COMMENTS

Reviewer #1 Comment 1: This contribution represents a detail HRTEM study of the structures of the tilt grain boundaries in nearly $\Sigma 5$ CSL orientation relationship documented by the APT measurements of the grain boundary chemistry in bcc-based iron system. As a starting material, the authors used a sequence of the samples cut of a curved grain boundary in a bicrystal. In this respect this work would represent an important contribution to understanding the structure/property relationship and hint for the grain boundary engineering approach. From this point of view, the paper represents an excellent contribution to the field.

Response: We are most grateful for the kind acknowledgment of our research and its significance in this field. Your positive feedback serves as a source of motivation for us to persist in exploring and advancing the frontiers of knowledge in this area.

Reviewer #1 Comment 2: Unfortunately, there are some problems which degrade the quality of the top experimental approach and do not allow me to recommend the manuscript for publication in the present form. There are two main reasons supporting my decision:

Response: We appreciate the reviewer's efforts in reviewing our manuscript and providing constructive feedback. We understand that there are some issues that have impacted the quality of our paper in its originally submitted form. In the following response, we fully comply and made all the necessary revisions to improve our manuscript and address the specific concerns raised by the reviewer.

Reviewer #1 Comment 2.1: I. The paper completely neglects the previous experimental results which were obtained earlier with the same conclusions despite more simple microscopic methods were used: In 1990s, the group of Lejcek studied (in numerous papers) the segregation of Si, P and C in bicrystals of the bcc Fe-Si based alloy with the conclusion showing that (310) and (210) symmetric boundaries and (001)//(011) asymmetric boundaries are singular. In fact, this is the same conclusion as done in the present manuscript. In a paper, Lejcek et al. did also use the same geometry of the sample as the authors in the present research. Nevertheless, no paper of this group (except the book (23) but in a quite different context) is cited in the manuscript.

Response: We sincerely appreciate the guidance from the reviewer in improving the quality of our manuscript. We thank the reviewer for bringing these references to our attention and we sincerely apologize for our mistake in not referencing the previous experimental results obtained by Lejcek and colleagues on the segregation of Si, P, and C in bicrystals of the bcc Fe-Si based alloy [Ref. 1-3, all references cited in our response letter have been listed at the end of the letter for the reviewer's convenience]. We have revised our manuscript now to include these multiple references to Lejcek's work and to acknowledge these profound contributions to this field accordingly.

Yet, we most respectfully tend to disagree with the reviewer's statement that our conclusion is the same as that of the previous work. Our study provides an in-depth atomic scale analysis and proof by enabling the imaging of the exact local atomic structures for grain boundaries with various

inclinations, the direct imaging of light solute atoms, and the two-dimensional mapping of grain boundary composition. Furthermore, our key conclusion is distinctly different from that of the previous excellent research in this field. We have demonstrated that the solute segregation behavior of grain boundaries is dependent on the specific atomic motif types and their sequential alignment. These atomistic structural building blocks affect the ability of grain boundaries to accommodate solutes. This result shows that the atomic motifs and their sequential combination that altogether constitute an interface play a key role in determining the overall solute segregation behavior of grain boundaries. In contrast, Lejcek et al.'s excellent and groundbreaking previous work rendered less direct atomic-scale observation that were capable of revealing both the atomic structure of the grain boundary motifs and their chemical composition at the same time.

Changes made in response to the comment:

Page 11, Line 14, Added: Lejcek and Hofmann have previously reported the anisotropic GB segregation behavior of P, Si, and C at various tilt GBs of $\Sigma = 5, 36.9^\circ [100]$ in an Fe-3.5at.%Si alloy using Auger electron spectroscopy^{40,41}. Special (singular) segregation behavior has been reported not only for symmetrical GBs, such as $\{013\}$ and $\{012\}$, but also for asymmetrical GBs, e.g., $(001)/(03\bar{4})$, $(0\bar{1}7)/(01\bar{1})$, and $(001)/(011)$ ⁴². The authors attribute the singular segregation behavior of these GBs to the high values of the interplanar spacing⁴¹.

Page 13, Line 27, Added: Through high-resolution atomic structure analysis and detailed chemical composition analysis, we discovered the strong dependence of the GB's solute segregation behavior on its specific underlying atomic motif types and their sequential alignment into a faceted arrangement. The resulting atomic structure affects the ability of GBs to accommodate solute atoms. With this we can trace a GB's segregation behavior back to the solubility and arrangement of its underlying atomic motifs. This changes our view of GB segregation from a top-down approach, where segregation has been associated with macroscopic kinematic degrees of freedom to a bottom-up approach, where the nature of the atomic motifs that are the underlying interfacial building blocks as well as their sequential arrangement determine a GB's solubility.

Reviewer #1 Comment 2.2: II. The interpretation of the GB segregation is vague.

As one can understand from the description in Methods, the samples were cut from the as-grown bicrystal. So, the reader understands that the segregation state cannot be ascribed to defined temperature. This is also supported by the width of the segregation layer (fig. 9/Extended data). It means that one cannot make ANY conclusion about the segregation behavior of individual grain boundaries! It is not clear whether the GB composition can correspond to low temperatures at which the segregation is high at general boundaries compared to singular GBs (= facets) or at high temperatures where the behavior is opposite? In addition, the composition of the grain boundaries in cooled as-grown bicrystal is out of any equilibrium, so no conclusion concerning the different behavior of individual GBs can be made from the composition of the boundaries.

Response: We thank the reviewer for the valuable feedback. We acknowledge that the previous description of the sample processing in our manuscript may have been insufficient. While we cited our previous publication in Nature Communication [Ref. 4], we understand that the lack of details on the fabrication of our bi-crystal may have caused confusion for the reviewer. In response to this, we fully comply to this point and we have decided to provide a more comprehensive and detailed description of the bi-crystal fabrication process, in addition to citing our previous work. We hope that this will address the issue, help avoid any misunderstandings, and improve the clarity of our manuscript.

Our bi-crystal was grown at a temperature of 1600 °C, with a rate of 2 mm/hour for a duration of 50 hours. Following the growth process, the oven was slowly cooled over a period of 24 hours at a rate of approximately 1.1 °C/minute. During this heat treatment, the temperature continued to decrease, and the segregation state was dependent on the overall history of the cooling process. In our previous work [Ref. 4], we developed a kinetic model that considers the time and temperature-dependent enrichment of solute in the grain boundaries. Based on this model and the heat treatment history, we determined that the kinetic limitation in terms of a lower threshold temperature for the segregation of B, Al, and C are 1200K, 800K, and 390K, respectively, below which the segregation was kinetically limited. We have also included a detailed description of this information in the revised manuscript.

We would like to highlight that the slow cooling rate of 1.1 °C/minute is a deliberate choice to minimize non-equilibrium segregation effects. This is in line with the findings described by Faulkner [Ref. 5]. Thus, we are confident that non-equilibrium segregation does not play a significant role in our case. Instead, we believe that the enrichment of solutes is predominantly caused by the binding energy of the solutes themselves with the grain boundary, which Faulkner refers to as equilibrium segregation [Ref. 5]. This distinction is important to note and has been included in the revised manuscript.

The reviewer has commented on the width of the segregation layer observed in our APT measurements, suggesting an unknown segregation behavior. However, we would like to clarify that the wide distribution of the segregation profile is mainly due to the local magnification artifact, which is a well-known phenomenon in the atom probe community [Ref. 6-7]. To obtain a more reliable quantification of the grain boundary composition, we also calculated the interfacial excesses, which are much less sensitive to the reconstruction artifact. We would like to highlight that the majority of interfacial excesses for both B and C at the investigated grain boundaries are less than 7.7 at./nm². This finding suggests that there is an adsorption of less than a monolayer of B or C atoms in the grain boundaries. Furthermore, we would like to draw the reviewer's attention to our DPC-4DSTEM reconstructions, in which we directly imaged the atomic columns of interstitial B and C atoms. These interstitial atoms only enriched the atomic motifs of grain boundaries. The direct imaging of interstitial atoms using our DPC-4DSTEM reconstructions has provided further confirmation of the narrow distribution of the solutes at grain boundaries.

The aim of this study is to conduct a comprehensive investigation of the structure-composition relationship for a series of grain boundaries that share the same misorientation but possess different inclination. Our primary objective is to gain a thorough understanding of the interplay between grain boundary structure and composition at the atomic or near-atomic resolution. The key conclusion of this study is that the decoration of grain boundaries is governed by atomic motifs, which exert significant influence over the most important chemical properties of grain boundaries. We attribute the diverse range of observed segregation states to the local variation of atomic motifs, which were not fully understood in previous work from the 1990s that utilized Auger electron spectroscopy with limited spatial resolution, which prevented the extraction of the atomic structure of grain boundaries [Ref. 1-3].

We appreciate the reviewer's comments on the potential role of temperature in influencing the grain boundary solute segregation. The reviewer has mentioned the potential existence of a "compensation temperature" [Ref. 8], at which a reversed anisotropy of grain boundary segregation may occur. To clarify any potential ambiguity, we have made revisions to the manuscript that include the specific temperature for the kinetic limit, below which the segregation of B, Al, and C was kinetically frozen in.

Our study provides new insights into the segregation phenomenon by utilizing high-resolution imaging to examine the structures of the grain boundaries with different inclinations. We found that these grain boundaries are formed by a limited number of atomic motifs. The sequential alignment of

these atomic motifs results in local facets that satisfy the required geometric constraints. Our observations suggest that each individual atomic motif exhibits a different solubility to accommodate interstitial solute atoms (as illustrated in Fig. 3 and Extended Data Fig. 6). The segregation of solute atoms, such as B, appears to be strongly correlated to the local facet structures formed by the sequential alignment of atomic motifs. While we agree that a more detailed temperature-dependent study would be an interesting avenue for future research, we believe that such an investigation would be beyond the scope of the current manuscript.

Changes made in response to the comment:

Page 19, Line 6, Added: The bi-crystal was grown at a temperature of 1600 °C, with a growth rate of 2 mm/hour for 50 hours. Following the growth process, the oven was slowly cooled down over a period of 24 hours at a rate of approximately 1.1 °C/minute. This choice of a slow cooling rate aimed to minimize non-equilibrium segregation effects^{24,43}. During the heat treatment, the temperature continued to decrease, and the resulting segregation state was dependent on the overall history of the cooling process. In our previous publication²⁴, we developed a kinetic model to determine the kinetic limitation in terms of a lower threshold temperature for the segregation of B, Al, and C, with 1200K, 800K, and 390K, respectively, below which the segregation was kinetically limited. The enrichment of solutes is predominantly caused by the binding energy of the solutes with the GB²⁴. Further details on the bicrystal synthesis and heat treatment can be found in our previous publication²⁴. It is worth mentioning that the wet chemical analysis of the bulk sample showed a content of 4 at.% Al, 0.001 at.% B, and 0.05 at.% C. The addition of Al serves to stabilize the bcc-Fe phase.

Page 37, Line 8, Added: The wide distribution observed in the segregation profiles of B and C, which can appear as thick as 10 nm, is primarily due to the local magnification artifact^{59,60}. Our DPC-4DSTEM results demonstrate that the interstitial atoms have primarily enriched the atomic motifs of the GBs, see Fig. 3 and Extended Data Fig. 6.

Page 14, Line 11, Added: Building upon the knowledge gained from this study, we plan to expand our investigation by exploring the temperature-dependent behavior of solute segregation and its influence on various mechanical and physical properties, such as corrosion, hydrogen embrittlement, and mechanical failure. These efforts will contribute to a more comprehensive understanding of the behavior of GBs and facilitate the development of more robust and reliable materials.

Reviewer #1 Comment 2.3: III. Further important comments:

Reviewer #1 Comment 2.3.1: 1. The grown bicrystal in this work contained two grains inclined by 38° around [001] axis and is ascribed to $\Sigma 5$ CSL orientation relationship. However, the true $\Sigma 5$ CSL misorientation is 36.9°[001]. If the study is done with the atomic resolution, the effect of the 1.1° deviation on the GB structure should be well explained. Very probably, this deviation can be explained by array of GB dislocations... Question is, however, how the complex structures are dependent on such deviation, e.g., that of the (11 1 0)/(980) GB?

Response: We appreciate the valuable insights provided by the reviewer. As correctly pointed out by the reviewer, our as-fabricated bicrystal deviates by 1.1° from the true $\Sigma 5$ CSL misorientation, which can potentially result in the formation of grain boundary defects or different structural units [Ref. 9-12]. The investigated grain boundaries in our study possess varying inclinations, which can also introduce a plethora of grain boundary defects such as disconnections, steps, or nano-facets. Both, misorientation and inclination can induce local atomic structural changes, which may significantly impact the grain boundary properties. To provide better clarity, we first discuss the influence of the

misorientation deviation on the local grain boundary structures of symmetric $\Sigma 5$ CSLs, followed by the discussion on the asymmetric ones.

The atomic structure of grain boundaries has been a subject of research for over half a century, and the structural unit model has been proposed to explain their structural features [Ref. 9-12]. For instance, the structural unit model can describe the structures of [001] symmetric tilt grain boundaries with misorientation angles ranging from 0 to 90° as a series of building blocks or structural units, labelled as "A", "B", and "C" units, as depicted in Fig. R1. This model represents the entire series of grain boundaries as a sequential alignment of individual structural units, each possessing a well-defined atomic structure known as the grain boundary core structure, rather than an array of dislocations. Further development of the structural unit model by Han et al. considered more metastable structures to match the true cases [Ref. 9]. In their work, they proposed that grain boundaries in the range of $36.87^\circ < \theta < 41.50^\circ$ misorientation are formed by "C" and "F" units, as illustrated in Fig. R2. Here, the "C" unit is the kite structure previously mentioned in our manuscript, and the "F" unit is equivalent to a grain boundary step. The existence of these steps, which are clearly visible in several HR-HAADF images for the #41 $(310) // (3\bar{1}0)$ grain boundaries shown in Fig. 2b and Extended Data Fig. 4b, is a result of the 1.1° deviation from the true $\Sigma 5$ misorientation of 36.87° .

Fig. R1 Grain boundary structures represented by "A", "B" and "C" units. Each structural unit is delineated by the dotted lines and outlined by the blue solid lines. The misorientation θ and the structural unit representation are indicated above each figure [Ref. 9].

Fig. R2 Grain boundary structures represented by "C" and "CF" units ("F" unit represents a step on the grain boundary). The misorientation θ and the structural unit representation are indicated above each figure [Ref. 9].

The local atomic configuration across the region of interest in the $(11\bar{1}0)//(980)$ asymmetric grain boundary is considerably more complicated. The Frank-Bilby equation [Ref. 13] suggests that long-range coherency strains resulting from the misorientation (deviated from the true $\Sigma 5$ boundary) and inclination can be canceled by Burgers vectors existing at different junctions of the faceted structure. Medlin et al. reported that an array of $(1/5)[310]$ and $(1/5)[120]$ dislocations can fully accommodate the misorientation and inclination from the true $\Sigma 5$ boundary [Ref. 14]. We have observed various

faceted structures for the $(11\ 1\ 0)//(980)$ grain boundary, including the commonly observed kite structure $(310) // (3\bar{1}0)$ and separated kite structure $(2\bar{1}0) // (1\bar{2}0)$, as well as other structures such as $(010) // (1\bar{1}0)$ and $(410) // (2\bar{1}0)$, as shown in Extended Data Fig. 4c. The dislocations that exist at the junction of the faceted structures play a key role in reducing the coherency strain between the two adjacent regions and stabilizing the grain boundary structure. We appreciated the suggestions from the reviewer and provide more discussion in the revised manuscript.

Changes made in response to the comment:

Page 6, Line 26, Added: The formation of this step is attributed to the fact that the current bicrystal deviates by 1.1° from the exact misorientation required for the perfect formation of a $\Sigma 5$ GB²⁹.

Page 6, Line 30, Added: The Frank-Bilby equation³⁰ suggests that long-range coherency strains resulting from the misorientation (1.1° deviation from the true $\Sigma 5$ GB) and inclination can be canceled out by Burgers vectors existing at different junctions of the faceted GB structures. For example, it has been reported by Medlin et al. that an array of $(1/5)[310]$ and $(1/5)[120]$ dislocations can fully accommodate the misorientation and inclination from the true $\Sigma 5$ GB¹⁷. The dislocations that exist at the junction of the faceted structures play a key role in reducing the coherency strain between the two adjacent regions and stabilizing the GB structure.

Reviewer #1 Comment 2.3.2: 2. When speaking about the nano-faceting, the representation $\Sigma 5$ $(010)//(110)$ is rather unlucky. I understand that it is the facet in $\Sigma 5$ orientation relationship but in fact, $(010)//(110)$ interface is incommensurable, i.e., non-CSL one as these two planes differ by $\sqrt{2}$ which is irrational number. I would recommend to use only $(010)//(110)$ without using ' $\Sigma 5$ '.

Response: We express our gratitude to the reviewer for thoroughly reviewing our manuscript and providing this valuable feedback item. In our paper, we intended to use the term ' $\Sigma 5$ ' to denote the orientation relationship of our bicrystal. However, the $(010)//(110)$ facet observed in our experiments mathematically contradicts the geometric requirement of $\Sigma 5$. The $(010)//(110)$ facet can only coexist with other grain boundary defects and facets to meet the orientation relationship of the $\Sigma 5$ grain boundary. Therefore, we understand that using ' $\Sigma 5$ ' in conjunction with $(010)//(110)$ may create confusion for the readers. We have decided to comply and use only $(010)//(110)$ without referring to ' $\Sigma 5$ ' to avoid any potential misunderstandings. The same principle has also been applied to other asymmetric grain boundaries. We appreciate the valuable input from the reviewer, which has helped us improve the clarity of our paper.

Changes made in response to the comment:

Page 6, Line 29, Modified to: ...such as $(310) // (3\bar{1}0)$, $(2\bar{1}0) // (1\bar{2}0)$, $(010) // (1\bar{1}0)$, etc....

Page 7, Line 3, Modified to: ...the majority of the local atomic motifs are the straight $(2\bar{1}0) // (1\bar{2}0)$ motifs with the intervening $(010) // (1\bar{1}0)$ facet...

Page 7, Line 5, Modified to: ...where three specific ones dominate, namely $\{110\} // \{010\}$, $\{310\} // \{310\}$, and $\{210\} // \{120\}$...

Page 7, Line 9, Modified to: ...However, there are less reports on the asymmetric $\{110\} // \{010\}$ atomic motifs...

Page 8, Line 7, Modified to: ...In Fig. 3a, we highlighted the $(110) // (010)$ atomic motif with a yellow triangle...

Page 8, Line 12, Modified to: ...indicating a higher segregation tendency than in the (110) // (010) atomic motif...

Page 9, Line 2, Modified to: ...Three representative local atomic motifs: **a** (110) // (010); **b** (310) // ($3\bar{1}0$); **c** ($2\bar{1}0$) // ($1\bar{2}0$)...

Page 10, Line 32, Modified to: ...The tetrahedral sites observed for the (110) // (010) atomic motif...

Page 23, Line 20, Modified to: ...Extended Data Fig. 5a shows an example of the reconstructed dark-field image containing (110) // (010) atomic motifs...

Page 24, Line 21, Modified to: ...Extended Data Fig. 5d presents the reconstruction of the (projected) electrostatic potential of the region containing (110) // (010) atomic motifs...

Page 25, Line 5, Modified to: ...Extended Data Fig. 5e shows the charge-density map for the (110) // (010) atomic motifs...

Page 25, Line 8, Modified to: ...More examples for three representative atomic motifs, (110) // (010), (310) // ($3\bar{1}0$), and ($2\bar{1}0$) // ($1\bar{2}0$)...

Page 25, Line 12, Modified to: ...Extended Data Fig. 6a i vs ii for the (110) // (010) atomic motif...

Page 33, Line 2, Modified to: ...Extended Data Fig. 5 | Experimental DPC-4DSTEM reconstruction for the (110) // (010) atomic motif...

Page 34, Line 2, Modified to: ...Extended Data Fig. 6 | Light solutes at $\Sigma 5$ GBs for three representative atomic motifs: **a i-ii** (110) // (010); **b i-iv** (310) // ($3\bar{1}0$); **c i-vi** ($2\bar{1}0$) // ($1\bar{2}0$)...

Reviewer #1 Comment 2.3.3: 3. L. 240: 'repulsive interaction' ... The behavior of B and C is definitively more complicated. First, there will be size competition as these two solutes are interstitial. Repulsion is definitively not excluded but I do not know the details. Additionally, the interaction of these solutes with Al can play a role.

Response: We appreciate the valuable input from the reviewer. The phrase "repulsive interaction" mentioned by the reviewer is more related to the theory study of the interaction between solute atoms. We did not use this phrase "repulsive interaction" in the manuscript. Instead, we use the term "mutually repulsive" to describe the observed phenomenon of the distribution of B and C content in the grain boundary. As stated in our manuscript, we find that the decoration features with B and C are not strongly correlated. However, certain regions appear to exhibit mutual repulsion, such that regions enriched in B tend to have a low content of C, and vice versa, as shown in Fig. 4b. We also note that the Pearson product-moment correlation coefficient between the local grain boundary compositions of B and C is weak, and the local content of B and C varies across the entire grain boundary plane. Our analysis suggests that the actual local composition of B and C strongly depends on various factors, including the atomic motif, defect density and nano-faceted structure. These findings underscore the importance of understanding the complex interplay between different factors in influencing the distribution of B and C in the grain boundary.

In our previous manuscript [Ref. 4], we developed a kinetic model to demonstrate that solute B diffuses faster into grain boundaries than solute C. The distribution of solute B and C atoms is influenced by both thermodynamics and kinetics. In the current manuscript, we referred to our previous work [Ref. 4] to explain the potential mechanism. Specifically, the negative segregation energy drives solute B to segregate into the first nearest interstitial site of the grain boundary. When

additional B or C atoms attempt to segregate into the second nearest interstitial sites, the segregation energy becomes positive, making further segregation energetically unfavorable. This discussion has been presented in our original manuscript (Page 10, Line 21). In our previous work [Ref. 4], we have also investigated the interaction of solute B and C with Al, and we found strong repulsive interactions between Al and either B or C, implying that co-segregation of these elements is highly unlikely.

In this study, we provide a more systematic experimental observation of the segregation tendency for solute B and C atoms as a function of local atomic motifs. Although more detailed theoretical work will be the goal of future studies, it is beyond the scope of this work.

Changes made in response to the comment:

Page 13, Line 27, Added: Through high-resolution atomic structure analysis and detailed chemical composition analysis, we discovered the strong dependence of the GB's solute segregation behavior on its specific underlying atomic motif types and their sequential alignment into a faceted arrangement. The resulting atomic structure affects the ability of GBs to accommodate solute atoms. With this we can trace a GB's segregation behavior back to the solubility and arrangement of its underlying atomic motifs. This changes our view of GB segregation from a top-down approach, where segregation has been associated with macroscopic kinematic degrees of freedom to a bottom-up approach, where the nature of the atomic motifs that are the underlying interfacial building blocks as well as their sequential arrangement determine a GB's solubility.

Reviewer #1 Comment 2.3.4: 4. L. 265: 'the GB inclination decreased from 60° for (310)//(310) to 0° for (210)//(210).' (310) and (210) symmetric GBs are inclined by 45° . Can you explain, what does mean these ' 60° '?

Response: We apologize for the error in our manuscript. The {310} and {210} symmetric grain boundaries are actually inclined by 45° , as indicated in Extended Data Fig. 1, where we provide the inclinations for all the relevant regions of interest. We have corrected the typo in the revised manuscript.

Changes made in response to the comment:

Page 11, Line 8, Modified to: ...the GB inclination decreased from 45° for $\Sigma 5$ (310)//($\bar{3}\bar{1}0$) to 0° for $\Sigma 5$ ($\bar{2}\bar{1}0$) // ($\bar{1}\bar{2}0$)...

Reviewer #2 Comment 1: The authors performed a holistic, multiscale analysis of the hierarchical, structural, and chemical features of GBs using high-resolution imaging techniques including HAADF-STEM, DPC-4DSTEM, and APT. They find that the chemical properties of GBs cannot be simply determined by their CSL value or mesoscopic angular features alone, but instead strongly depend on their GB inclination, the resulting faceting, and particularly on the local atomic motifs. Some suggestions are shown as follow:

Response: We are grateful for the nice summary of our work!

Reviewer #2 Comment 1.1: 1. The authors exploit the strong interdependence of the interface structure and chemistry and find that the change of the inclination of the grain boundary plane with

identical misorientation impacts grain boundary composition and atomic arrangement. What is the relationship between the inclination of GB, type of GB and GB compositions.

Response: The reviewer correctly noted that our research examined how grain boundary inclination impacts its structure and composition. Even though we strictly limit the geometric freedom of grain boundaries, their atomic arrangement remains extremely intricate. The grain boundaries exhibit a wide range of nano-faceted structures and defects for various grain boundary inclinations. Medlin et al. reported that the misorientation and inclination of a $\Sigma 5$ grain boundary can be accommodated by an array of $(1/5)[310]$ and $(1/5)[120]$ dislocations [Ref. 14]. However, in reality, it's more complex than that. We also discovered many other atomic motifs, such as the $(110)//(001)$ type.

When studying interface chemistry, we do not provide just a single value for grain boundary composition, Instead, we create two-dimensional maps that show how the composition varies locally along the grain boundary plane. The variation in grain boundary composition arises from atomic motif types and the nano-faceted structure observed for the different types and degrees of grain boundary plane inclination. We had picked and presented three typical atomic motifs for direct visualization of light solute atoms B and C, as shown in Fig. 3 and Extended Data Fig. 6. The $(110)//(010)$ atomic motif has fewer interstitial sites, while the $(2\bar{1}0) // (1\bar{2}0)$ motif can accommodate more solute atoms by varying and accommodating its local motif structure and the associated distortions. The $(310) // (3\bar{1}0)$ atomic motifs are in-between these two types, specifically regarding their solubility to accommodate and host solute atoms. The atomic motif types and their sequential alignment play a key role in controlling the chemical composition of grain boundaries. For instance, the grain boundary with an inclination of 60° is predominantly formed by low solubility atomic motifs, such as $(110) // (010)$ and $(310) // (3\bar{1}0)$, which translates to the lowest overall solute content for this specific grain boundary. Since the grain boundary with 0° plane inclination contains a high fraction of $(2\bar{1}0) // (1\bar{2}0)$ type atomic motifs, it also exhibits the highest overall solute content. The observed dependence of solute segregation behavior on grain boundary inclination is related to specific atomic motif types and their sequential alignment into a faceted arrangement. These structural features affect the solubility of grain boundaries to accommodate solute atoms, suggesting that the atomic motifs play the key role in determining the overall solute segregation behavior of grain boundaries.

Changes made in response to the comment:

Page 3, Line 20, Added: Analogous to building blocks or Lego bricks, a limited number of atomic motifs with their sequential alignment form complex GB structures with a wide range of microscopic degrees of freedom and varying solubility for accommodating solute atoms.

Page 13, Line 16, Added: The variation in GB composition arises from atomic motif types and the nano-faceted structure observed for the different types and degrees of GB plane inclination. We had picked and presented three typical atomic motifs for direct visualization of light solute atoms B and C, as shown in Fig. 3 and Extended Data Fig. 6. The $(110)//(010)$ atomic motif has fewer interstitial sites, while the $(2\bar{1}0) // (1\bar{2}0)$ motif can accommodate more solute atoms by varying and accommodating its local motif structure and the associated distortions. The $(310) // (3\bar{1}0)$ atomic motifs are in-between these two types, specifically regarding their solubility to accommodate and host solute atoms. Combining these atomic motifs along the GB plane in the form of a motif sequence thus leads to variations in the local GB chemical composition. For instance, the GB with an inclination of 60° is primarily formed by low solubility atomic motifs, such as $(110) // (010)$ and $(310) // (3\bar{1}0)$, which translates to the lowest overall solute content for this specific GB. Since the GB with 0° plane inclination contains a high fraction of $(2\bar{1}0) // (1\bar{2}0)$ type atomic motifs, it also exhibits the highest overall solute content. Through high-resolution atomic structure analysis and detailed chemical composition analysis, we discovered the strong dependence of the GB's solute segregation behavior on its specific underlying atomic motif types and their sequential alignment into a faceted

arrangement. The resulting atomic structure affects the ability of GBs to accommodate solute atoms. With this we can trace a GB's segregation behavior back to the solubility and arrangement of its underlying atomic motifs. This changes our view of GB segregation from a top-down approach, where segregation has been associated with macroscopic kinematic degrees of freedom to a bottom-up approach, where the nature of the atomic motifs that are the underlying interfacial building blocks as well as their sequential arrangement determine a GB's solubility.

Reviewer #2 Comment 1.2: 2. I was expected more investigations on this topic. The authors have shown the microstructure of GBs including the inclinations of GB, GB composition, and atomic arrangement, but how these elements influence the mechanical and transport properties of materials (such as corrosion, hydrogen embrittlement, or mechanical failure) is more important? Without this part of work, this manuscript looks uncompleted.

Response: We appreciate the reviewer's feedback. Our manuscript aims to establish connections between grain boundary geometry, atomic structure, and composition. We conducted a thorough characterization of the structure and composition of various grain boundary structures, which helped us understand how atomic motifs govern the decoration of grain boundaries. In addition to conventional HR-HAADF and APT measurements, we developed a new analysis method for direct imaging of light solute atoms. This method allows for simultaneous determination of the positions of both heavy elements like Fe and lighter elements such as B and C. We successfully imaged B atoms in the grain boundary of iron for the first time and identified them as interstitial atoms. This discovery will be valuable for theoreticians to develop and verify their atomic models of grain boundaries. We believe that our findings will be of interest to the general readership of Nature Communication.

We agree that measuring mechanical and transport properties is crucial. These properties are strongly related to the grain boundary structure and composition. A deep understanding of the structure-composition relationship is an essential preliminary step for any further in-depth study. As it can be seen for the reviewer, the current finding is already very rich. We plan to focus on the structure-composition relationship for this work and investigate their relationship with properties in our future research. Currently, we are studying the corrosion behavior of grain boundaries with Al, B, and C. We have found that the corroded grain boundary forms a thick oxidation layer (approximately ten nanometers), but the original segregated B and C atoms do not segregate further. While this finding is interesting, it does not contribute significantly to the current storyline of this manuscript. The current work focused on the structure and composition of the atomic motifs at grain boundaries. We believe that the length scales of typical corrosion and mechanical experiments may not be appropriate for studying individual facets, such as $(310) // (3\bar{1}0)$, $(2\bar{1}0) // (1\bar{2}0)$ or $(110) // (010)$ atomic motifs. In reality, the nano-faceted structure can easily form. Even experiments that are downscaled to micron-sized dimensions may only provide an integral answer to the properties of atomic motifs [Ref. 15].

We have also investigated the liquid metal embrittlement of Zn for these grain boundaries. We found that pre-segregated B can reduce the segregation of Zn, thereby mitigating grain boundary embrittlement. This paper has recently been accepted in Advanced Materials [Ref. 16].

We have included more information on our future research plans in the revised manuscript.

Changes made in response to the comment:

Page 14, Line 11, Added: Building upon the knowledge gained from this study, we plan to expand our investigation by exploring the temperature-dependent behavior of solute segregation and its influence on various mechanical and physical properties, such as corrosion, hydrogen embrittlement,

and mechanical failure. These efforts will contribute to a more comprehensive understanding of the behavior of GBs and facilitate the development of more robust and reliable materials.

Reviewer #2 Comment 1.3: 3. Fig. 2 can be divided into two figures (a-e for fig.2 for Facet structures discussion, f-h for fig.3 for Imaging light B and C atoms at atomic motifs discussion).

Response: We have followed the reviewer's suggestion and divided Fig. 2 into two separate figures in the revised manuscript.

Changes made in response to the comment:

Page 7, Line 11, Modified to: ...Fig. 2 | Local faceted GB structures...

Page 9, Line 2, Modified to: ...Fig. 3 | Imaging light B and C atoms at atomic motifs. Three representative local atomic motifs: **a (110) // (010); **b** (310) // (3 $\bar{1}$ 0); **c** (2 $\bar{1}$ 0) // (1 $\bar{2}$ 0)...**

Page 12, Line 2, Modified to: ...Fig. 4 | Local chemical composition analysis for Σ 5 GBs...

References

- Ref. 1. P. Lejček and S. Hofmann, *Acta Metallurgica et Materialia* 39, 2469 (1991).
- Ref. 2. P. Lejček et al., *Surf. Sci.* 264, 449 (1992).
- Ref. 3. P. Lejček and S. Hofmann, *Surf. Sci.* 307-309, 798 (1994).
- Ref. 4. A. Ahmadian et al., *Nat. Commun.* 12, 6008 (2021).
- Ref. 5. R. G. Faulkner, *J. Mater. Sci.* 16, 373 (1981).
- Ref. 6. M. Miller and M. Hetherington, *Surf. Sci.* 246, 442 (1991).
- Ref. 7. F. Vurpillot et al., *Appl. Phys. Lett.* 76, 3127 (2000).
- Ref. 8. P. Lejček and S. Hofmann, *Interface Sci.* 9, 221 (2001).
- Ref. 9. J. Han et al., *Acta Mater* 133, 186 (2017).
- Ref. 10. A. P. Sutton et al., *Philos. Trans. Royal Soc. A* 309, 1 (1983).
- Ref. 11. A. P. Sutton and V. Vitek, *Philos. Trans. Royal Soc. A* 309, 37 (1983).
- Ref. 12. A. P. Sutton et al., *Philos. Trans. Royal Soc. A* 309, 55 (1983).
- Ref. 13. A. P. Sutton and R. W. Balluffi, *Interfaces in Crystalline Materials* (Clarendon Press, 1995).
- Ref. 14. D. L. Medlin et al., *Acta Mater* 124, 383 (2017).
- Ref. 15. H. Bishara et al., *ACS Nano* 15, 16607 (2021).
- Ref. 16. Ahmadian et al., *Adv. Mater.* n/a, 2211796 (2023).

REVIEWERS' COMMENTS

Reviewer #1 (Remarks to the Author):

The authors did consider my comments and responded all of them in the way that I have no more comments to it.

Reviewer #2 (Remarks to the Author):

My comments are well addressed in the response.